# Assessing the Impacts of Land Use and Climate Changes on River Discharge towards Lake Victoria

Renatus James Shinhu [1,2], Aloyce I. Amasi [2,*], Maarten Wynants [3], Joel Nobert [4], Kelvin M. Mtei [2] and Karoli N. Njau [2]

1 Lake Victoria Basin Water Board (LVBWB), Mwanza P.O. Box 1342, Tanzania
2 School of Material, Energy, Water and Environmental Science, The Nelson Mandela African Institution of Science and Technology, Arusha P.O. Box 447, Tanzania
3 Department of Soil and Environment, Swedish University of Agricultural Sciences, Lennart Hjelms väg 9, 75007 Uppsala, Sweden
4 Institute of Resource Assessment, University of Dar es Salaam, Dar es Salaam P.O. Box 35095, Tanzania
* Correspondence: aloyce.amasi@nm-aist.ac.tz; Tel.: +255-766-877-920

**Abstract:** The Lake Victoria basin's expanding population is heavily reliant on rainfall and river flow to meet their water needs, making them extremely vulnerable to changes in climate and land use. To develop adaptation and mitigation strategies to climate changes it is urgently necessary to evaluate the impacts of climate change on the quantity of water in the rivers that drain into Lake Victoria. In this study, the semi-distributed hydrological SWAT model was used to evaluate the impact of current land use and climate changes for the period of 1990–2019 and assess the probable future impacts of climate changes in the near future (2030–2060) on the Simiyu river discharge draining into Lake Victoria, Northern Tanzania. The General Circulation Model under RCPs 4.5, 6.0 and 8.5 predicted an increase in the annual average temperature of 1.4 °C in 2030 to 2 °C in 2060 and an average of 7.8% reduction in rainfall in the catchment. The simulated river discharge from the hydrological model under RCPs 4.5, 6.0 and 8.5 revealed a decreasing trend in annual average discharge by 1.6 m$^3$/s from 5.66 m$^3$/s in 2019 to 4.0 m$^3$/s in 2060. The increase in evapotranspiration caused by the temperature increase is primarily responsible for the decrease in river discharge. The model also forecasts an increase in extreme discharge events, from a range between 32.1 and 232.8 m$^3$/s in 1990–2019 to a range between 10.9 and 451.3 m$^3$/s in the 2030–2060 period. The present combined impacts of climate and land use changes showed higher effects on peak discharge at different return periods (Q5 to Q100) with values of 213.7 m$^3$/s (Q5), 310.2 m$^3$/s (Q25) and 400.4 m$^3$/s (Q100) compared to the contributions of climate-change-only scenario with peak discharges of 212.1 m$^3$/s (Q5), 300.2 m$^3$/s (Q25) and 390.2 m$^3$/s (Q100), and land use change only with peak discharges of 295.5 m$^3$/s (Q5), 207.1 m$^3$/s Q25) and 367.3 m$^3$/s (Q100). However, the contribution ratio of climate change was larger than for land use change. The SWAT model proved to be a useful tool for forecasting river discharge in complex semi-arid catchments draining towards Lake Victoria. These findings highlight the need for catchment-wide water management plans in the Lake Victoria Basin.

**Keywords:** SWAT; hydrological model; Simiyu catchment; river discharge; water quantity; agricultural land; deforestation





## 1. Introduction

Catchment topography, soil type, and climatic and vegetation dynamics are the major variables that affect the hydrology of a watershed [1–4]. Vegetation is a major component of terrestrial ecosystems as a primary producer of organic material, and thus plays an important role in energy flow, the water cycle and buffering against desertification [5]. In addition, terrestrial vegetation offers regulatory services that include flood attenuation and nutrient retention, soil erosion control, climate regulation and enhancing river

discharge [6–11]. Permanent vegetation and soil organic matter intercept rainfall, promote infiltration and protect the soil surface from evaporation. When those two factors are reduced, less rainfall will infiltrate the soil, leading to more runoff over the surface. However, recent observed increased flashiness in river discharges around the globe is mostly attributed to anthropogenic factors such as land cover changes and global climatic changes, which have caused changes in the hydrologic cycle and increases in the catchment's climate [12]. Recent advances in remote sensing data have shown great potential for studying land use land cover changes threatening catchments' ecosystem functions and services [13–16]. Remote sensing techniques allow for repeatable imaging of the same area needed to determine changes and temporal patterns in catchment ecosystems [14–16]. However, in semi-arid environments, precipitation and evapotranspiration vary seasonally with climate variability and anthropogenic activities, making the Soil and Water Assessment Tool (SWAT) an ideal choice for this study due to its proven performance under these conditions [17,18].

In the Lake Victoria Basin in Northern Tanzania, an increasing demand for food and energy from a growing population has led to an expanse in agricultural areas and widespread deforestation [19,20]. The pressure has even led to alterations in the catchment's agricultural practices, shifting from traditional mixed perennial agriculture toward more intensive cropping with minimal devotion to soil and water conservation [21,22].

Recent expansions of gully networks have also increased the connectivity from hillslopes to the river networks, leading to increased drainage rates [23]. This higher hydrological connectivity generally leads to bigger and more rapid differences between the peak and base flows, which increases the risks of both floods and droughts [24–26]. The dynamics of surface runoff, evapotranspiration, infiltration, groundwater flow, and river discharge are significantly impacted by poor land and water management associated with unsustainable agriculture and deforestation in the catchments [24,27,28]. Therefore, it is important to understand how present and future climate variability and land use change will influence the discharge of the rivers draining towards Lake Victoria. The Simiyu River contributes on average 16 m$^3$/s to Lake Victoria and thereby is fifth largest in terms of water flow, behind Kagera (225 m$^3$/s), Mara (50 m$^3$/s), Ngono (27 m$^3$/s) and Grumeti (20 m$^3$/s) Rivers [24]. The catchment is of significant interest to national, regional and international reputation as it is part of the Nile basin, which is shared by 11 riparian countries and Lake Victoria. Additionally, it is crucial for maintaining the country's economy, supporting local communities, and for biodiversity conservation. The catchment's major economic activities include rainfed smallholder agriculture, fishing and pastoralism [29]. The Simiyu river provides water for both domestic and industrial uses in some towns around Mwanza Region. The river and its tributaries are also targeted for supply of additional water for irrigation in the catchment. The water demands in the catchment are rapidly growing and are anticipated to increase even further in the future [29]. However, due to erosion, inadequate human waste management, unsustainable land use management practices, degradation of wetlands, and sand mining in the river, the catchment's water resources are declining in both quality and quantity [26,27]. Increasing agricultural activity makes catchments more susceptible to both floods and droughts by increasing runoff and decreasing infiltration [30]. The Simiyu catchment has experienced flooding in the last decades, which has impacted both human safety and agricultural yields [31]. In 2007, the study area received intensive rains that culminated into floods, eventually resulting in the injury, damage and loss of life and assets, local infrastructure distraction, and forced relocation [20,32,33]. Studies in the area assume that the increased flood peaks are caused by increasing settlements that have led to deforestation, wetland degradation and increased impervious surfaces [34,35]. However, there is little to no empirical evidence about how the land use and climatic variability affect the current discharge, nor do we know how future climate change will affect this either. Dynamic catchment models can be calibrated using existing hydrological data and subsequently applied to reconstruct and forecast river discharges [36,37]. They are founded on the water balance equation of the basic water cycle elements (such as precipitation,

infiltration, evapotranspiration) and the physical catchment features that influence runoff, such as topography, soil type and land use [38]. Locally calibrated catchment models can be instrumental for simulating the impacts of land use change, soil management and climate change on the flow of water and nutrients. The catchment's semi-arid environment, distinct rainfall seasonality and climate conditions made the Soil and Water Assessment Tool (SWAT) an ideal choice for this study because of its demonstrated performance in such conditions [17,18]. The model is semi-distributed and spatially referenced to a specific catchment or sub-catchment where the smallest defined sub-catchments are routed together using the stream network. The sub-catchments are built on hydrological response units (HRUs), which are groups of similar land uses, soils, and slopes within the sub-catchment. Understanding the detailed spatially explicit datasets of HRUs of watersheds to the signals of physical (land use) and climatic (rainfall and air temperature) variables is thus an important component of water resources planning management. Therefore, this study aimed to evaluate the impacts of climate and land use change on the streamflow at the critical agro-ecological region of the Simiyu catchment. The findings from this study will give insights for developing and implementing adaptation and mitigation measures to minimize the impacts of climate change and land use on water resources for sustainable economic development.

## 2. Materials and Methods

### 2.1. Description of the Study Site

The Simiyu River catchment is situated in the Simiyu and Mwanza regions between Itilima, Bariadi, Busega and Magu Districts in the Lake Victoria Basin (LVB) (Figure 1). The catchment covers an area of circa 11,000 km$^2$ and drains towards Lake Victoria. The catchment extends between latitude 2°15′ and 3°15′ South and longitudes 33°15′ and 35°00′ East, and its altitude ranges between 1100 and 2000 m.a.s.l. The catchment experiences semi-arid environments, with periodic variations in rainfall, most of which falls during the two rainy seasons of extended rains from March to May and short rains from October to December [29]. Between June and September there is a dry season, and January and February serve as the transitional months between the seasons [29,39]. The region also has significant interannual variation in rainfall with drier and wetter years. The mean annual rainfall in the catchment varies spatially as well, with the lower parts receiving an average of 750 mm annually and upper parts receiving an average of 1100 mm annually [29] (Supplementary Materials S1). The trend of rainfall is shown in the Supplementary Materials S1 for a period between 1990 to 2019. The natural vegetation of the area follows the altitudinal rainfall gradient, but is also impacted by human activities and downstream flood regimes. As a result, the catchment has nine (9) major land use and land cover types: grassland, forest, bush land, cultivated land, urban areas, woodland, water bodies, wetland, and bareland. The catchment is mostly underlined by granitoids, migmatite, mafic and ultramafics and meta-sediments, while fine-coarse clastic sediments, mafic volcanics, meta-basalts, phyllite, volcanic ashes, tephra, calcareous tuffs and sandy, gravelly, silty sediments represent smaller portions of the catchment [39] (Supplementary Materials S2). The map (Supplementary Materials S3) displays the study area's soil classes, where the major soil types are sandy clay (1.9%), loam (2.9%), clay (5%), clay loam (12.9%), sandy clay loam (13.5%) and sandy loam (63.8%) [26,29].

### 2.2. Data Preparation

SWAT requires spatial data, hydrological data and meteorological data for, respectively, building, calibrating, and forcing the model. The required input data were collected from different sources and prepared in the ArcGIS 10.2.1 environment to acquire the necessary setup essential for the ArcSWAT12 database. A digital elevation model (DEM) of the Simiyu catchment with a resolution of 1 arc second (30 m × 30 m) was included in the spatial data and was acquired from the United States Geological Surveys (USGS) website (http://gdex.cr.usgs.gov/gdex/ accessed on 28 March 2020), and was used to

delineate the watershed and the stream networks following the procedure in [17,40]. The land use/cover map of 1990 and 2019 (Figure 2a and b, respectively) were downloaded from Earth Explorer in May 1990 with Landsat 5 (resolution 30 m) and Landsat 8 images (resolution 30 m) captured in May 2019 and interpretation, atmospheric correction and geometric rectification performed using impact toolbox software (http://glovis.usgs.gov/ accessed on 17 December 2021). Geotagged photographs and field notes were collected from numerous ground observation operations (through field surveys and interviews with local people) to ensure full documentation of the land cover spectrum. Using these ground observations, supplemented by Google Earth imagery, the main land cover types in the region were mapped into a spectral signature file developed from training samples. The supervised classification according to ArchMap's maximum likelihood algorithm method uses these signature files to estimate predefined land cover classes from the entire Landsat image database. Visual inspection and comparison with high-resolution aerial images provided by Google Earth was used to remove potentially misclassified features. A raster calculator function was used to determine the appropriate elevation for specific land use classes based on expert knowledge of the study area as detailed by Taweesuk et al. [41]. Expert classification aims to improve the classification accuracy used to combine remote sensing data with other sources of georeferencing information such as digital elevation models (DEMs), land use and spatial texture data. The accuracy assessment was performed to determine the level of agreement between classified images and ground features. Overall accuracy ratings for images observed in 1990 and 2019 were 87.37% and 85.74%, respectively. This value meets the minimum accuracy threshold of 85% required for effective and realistic land use/cover change analysis and modeling [42,43]. The results of this study are considered acceptable because the accuracy values are greater than 80%, as reported by Jensen [44].

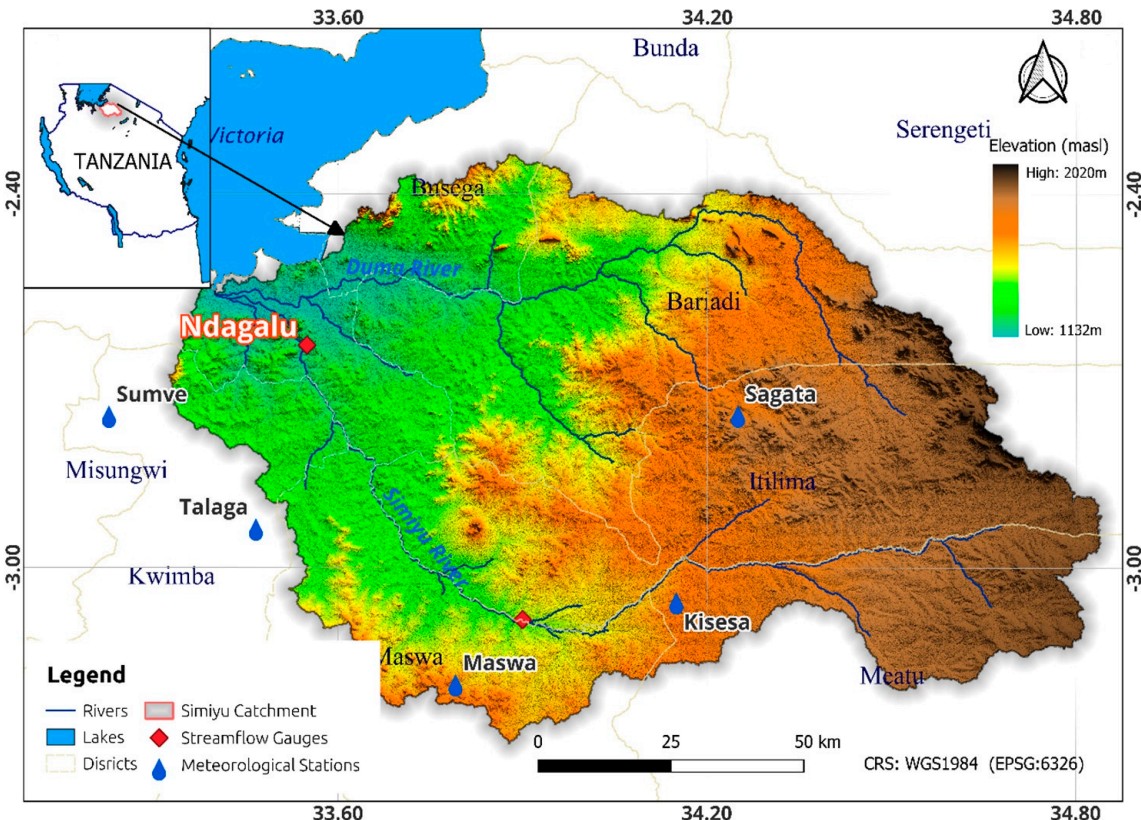

**Figure 1.** Location of the Simiyu River catchment detailing the elevations, rainfall and hydrometric stations.

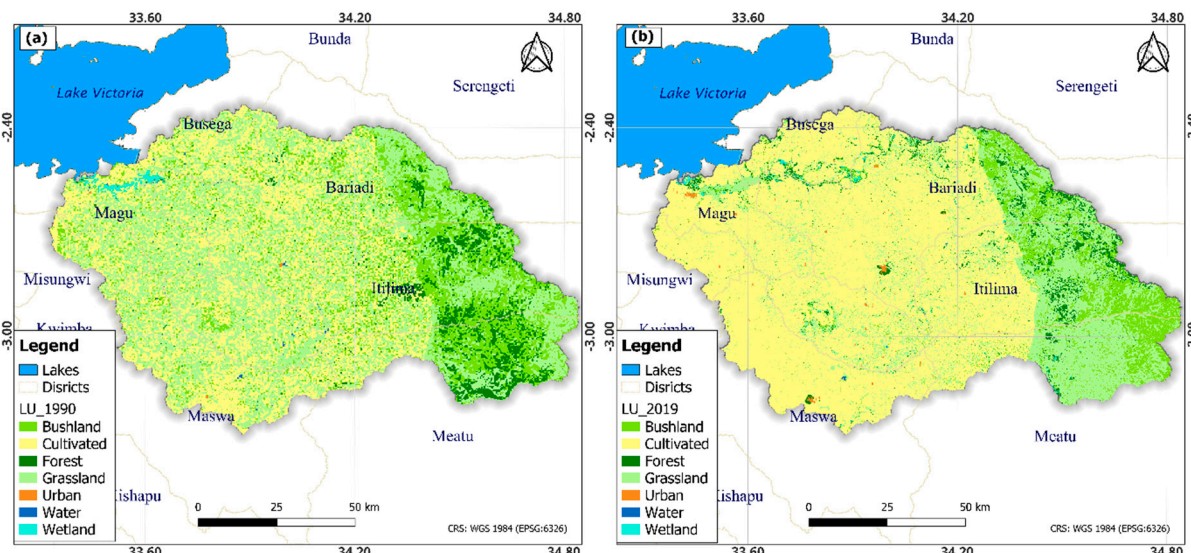

**Figure 2.** Land cover maps derived from Landsat imagery of (**a**) 1990 and (**b**) 2019, detailing the changes in land use land cover from 1990 to 2019.

The Lake Victoria Basin Water Board (LVBWB) provided the meteorological data such as daily rainfall (rainfall records for 5 stations Maswa, Sumve, Talaga, Sagata and Kisesa) and minimum and maximum temperature [40] (Table 1), supplemented by data from Tanzania Meteorological Agency (TMA) (satellite data from the Earth System Grid Federation (ESGF)). The hydrological data included the daily river discharge records from Ndagalu gauging station (−2.65299° S, 33.541930° E) between 1 January 1972 and 31 December 1996. The soil data were obtained from FAO Harmonized global soils database at (http://www.waterbase.org/download_data.html accessed on 17 December 2021) Digital Soil Map of the World (DSMW) [45] (Supplementary Materials S4). However, there were missing data in the rainfall patterns that could hide true patterns in the data and impede the analysis and interpretation of the flow variability results in complexity and uncertainty in modelling. Encountering data gaps is unavoidable, particularly in developing countries, hence various methods for handling infilling of missing data have been developed. In this study, filling missing data was performed using the RClimtool software version 1.

**Table 1.** Meteorological stations in the Simiyu catchment.

| Station ID | Meteorological Stations | Latitude ° S | Longitude ° E | Elevation | Daily Rainfall | Percentage Missing (NA%) |
|---|---|---|---|---|---|---|
| 933305 | Maswa | −3.182 | 33.79098 | 1334 | 1971–2019 | 7.182 |
| 923301 | Sumve | −2.751 | 33.2265 | 1243 | 1971–2019 | 30.814 |
| 923240 | Talaga | −2.932 | 33.46581 | 1237 | 1971–2019 | 0.691 |
| 923401 | Sagata | −2.75 | 34.25 | 1394 | 1971–2019 | 17.227 |
| 933406 | Kisesa | −3.05 | 34.15 | 1343 | 1971–2019 | 45.915 |

*2.3. SWAT Model Setup, Sensitivity Analysis, Calibration and Validation*

There are two main categories of statistical evaluations used to assess the performance of the best parameter sets chosen in the sensitivity analysis, i.e., model performance evaluation and uncertainty in model predictions. The statistical analysis parameters proposed by [46], such as the Nash–Sutcliffe efficient (NSE), a ratio of the root mean square error to the standard deviation of measured data (RSR) and the percentage bias (PBIAS), were used to assess the model performance in predicting the catchment conditions [47]. The *r*-factor and the *p*-factor were used for model prediction uncertainty [18]. The *p*-factor is the percentage of observations covered by the 95% prediction uncertainty, while the *r*-factor refers to the thickness of the 95% prediction uncertainty (95PPU) envelope. The

*p*-factor value ideally falls between 0 and 100%, while the *r*-factor falls between 0% and infinity [48]. An exact simulation of the measured data has a *p*-factor of 1 and *r*-factor of zero [49]. The extent to which model results deviate from these values can be used to assess the model's representativeness and the need for further calibration. A *p*-factor value of >70% and *r*-factor value of around 1 are suggested in semi-arid regions [50].

In this study, the catchment's hydrological responses to land use and climate changes were quantified using the climate scenario and the annual runoff coefficients of each land use. This enables the evaluation of the water resource dynamics, which are controlled by the succession of wet and dry years in the studied catchment. The multi-decadal climate prediction was analyzed in accordance with [51] using 30-year average annual and monthly results to obtain river discharge predictions for reference and future scenario periods. All the elements of the water balance in the study catchment were estimated using the hydrological component of the SWAT model. The model was built by partitioning the catchment into sub-catchments that are composed of several HRUs with relatively uniform combinations of land use/land cover, soil types, and topography. It is assumed that each HRU has similar hydrological processes [18,40,52]. The required climatic driving variables (daily rainfall, minimum and maximum temperature) were subsequently fed into the model, consequently determining the evapotranspiration rate by using the Hargreaves method [53]. After all the above processes were completed, the model was calibrated, validated, and assessed for performance accuracy and efficiency. Validation was carried out by using the split sample test whereby two time periods were selected for this analysis [54]: a calibration period of 1972–1982 and a validation period of 1988–1992. The daily river discharge data of 1972 to 1982 and 1988 to 1992 from the Ndagalu gauging station were used to calibrate and validate the SWAT model, respectively. SWATCUP SUFI-2, a semi-automatic calibration and uncertainty program, was used for the calibration and validation [18]. Model initialization was carried out during model calibration over a four-year warm-up period, from 1988 to 1992. Validation used the same number of calibration iterations as before; however, the sensitivity analysis was first performed then followed by the calibration process. The number of iterations used for calibration was maintained for validation [55]. Sensitivity analysis, according to [56], entails identifying the parameters that are most sensitive for a given basin and calculating the rate at which model outputs change as a result of changing model inputs. Sensitivity analysis was performed in ArcSWAT with and without discharge data from gauging stations and in SWAT-CUP using the SUFI procedures by running the model 1000 times. To rank the sensitivity of the parameters, the *t*-statistic and *p*-values are used. The most sensitive parameters are those with the lowest *p*-value and the highest absolute value of the *t*-stat. For the details on the whole calibration and validation of the model, readers can be referred to [49,50].

### 2.4. Calibration and Validation of Future Climate Data

General Circulation Models (GCMs) are useful for describing and forecasting future climate change patterns. In this study, the 1990–2019 period was used as the baseline and the near-future (2030–2060) was accounted for in climatic projections. Considering expansive numbers of accessible climate models and computational and human resource constraints, detailed climate change impact studies cannot incorporate all projections. In practice, one climate model or a small ensemble of climate models that covers more climatological variables using cluster analysis algorithms [57,58] is usually selected for the assessment based on their skill to simulate the present and near-future climate [59,60]. In addition, the poor temporal and coarse spatial resolutions of GCM outputs (usually precipitation and temperature) might be biased, limiting the effectiveness of GCM model outputs in providing useful information at the regional scale [61], and are thus downscaled to convert GCM outputs into regional high-resolution meteorological fields required for reliable hydrological modelling of particular catchments. For this study, 4 climate models from the Coupled Model Intercomparison Project 5 (CMIP5), i.e., CMCC.CM, ACCESS 1.3, MIROC5 and CNRM.CM, were downscaled and compared to find the most

representative of the Simiyu catchment climatological patterns and spatial variations [62] (Supplementary Materials S5). The three (3) climate models (MIROC5, CNRM.CM5 and ACCESS 1.3) replicated climate variability of the Simiyu catchment (Northen Tanzania) with high accuracy coefficient correlations of 0.96, 0.97 and 0.98, respectively (Figure 3); thus, 3 climate models were assumed to be the most representative in simulating spatial patterns in the decadal change of climate zones [63] in the catchment (Supplementary Materials S5). Forecasting climate change impacts on water resources is cumbersome [64] and requires using viable scenario changes detailed by the Intergovernmental Panel on Climate Change (IPCC) [64]. As shown in the Supplementary Materials S6, the IPPC's fifth assessment report from 2014 presented four Representative Concentration Pathways (RCPs) emission scenarios: RCP 2.6 (low emission scenario), RCP 4.5 (low–medium emission scenario), RCP 6.0 (medium–high emission scenario) and 8.5 (high emission scenario). In this study, three RCPs (4.5, 6.0 and 8.5) were used to analyze the future (2030–2060) climate change impacts because they assume an increase in GHG emissions until 2080, followed by a decline [65]. The steps outlined in the Guide for Running AgMIP Climate Scenario Generation Tools with R were used to create the near-future climate scenario of precipitation and temperatures [24,65,66]. The RCPs 4.5, 6.0 and 8.5 with ensembled GCM (ACCESS1.3, MIROC5 and CNRM.CM5) models were subsequently downscaled to the watershed level [67] using the Simple Delta Method, as it retains the historical patterns of the gridded observations [24]. In order to statistically downscale the selected models, the delta change algorithm that was acquired [64], along with the CMIP5-GCMs, was used to calculate the change factor or the ratio between a mean value in the future and historical run [64]. In order to create a time series that represents the future climate, this change factor was then applied to the observed time series 2030–2060 [64,68]. The downscaled and selected 3 GCMs climatic data points under RCPs 4.5, 6.0, and 8.5 were used as forcing data to forecast the river discharge under a future climate.

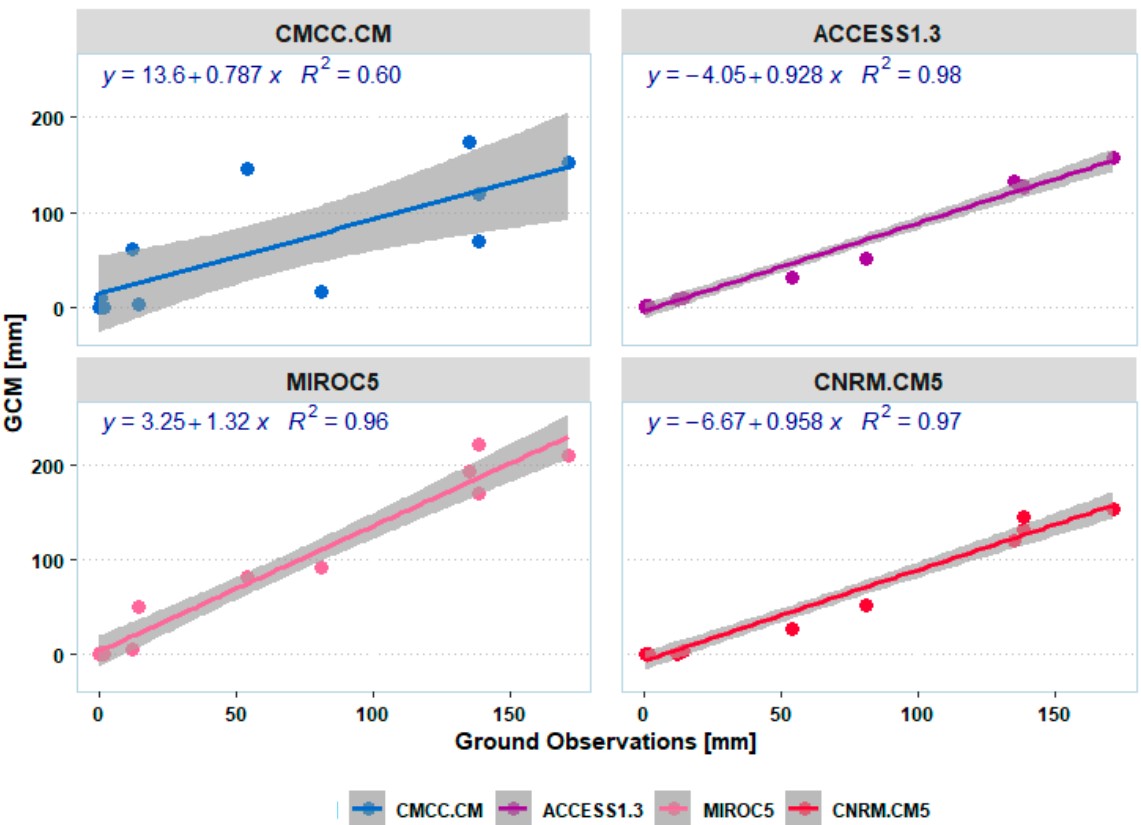

**Figure 3.** Correlation of the selected GCMs models used in this study (https://esgf-node.llnl.gov/search/cmip5/ accessed on 18 December 2021).

## 2.5. Simulating the Impacts of Land Use and Climate Change on Stream Discharge

The baseline land use/cover of 1990 was replaced in the calibrated SWAT model with the 2019 land use/cover of the Simiyu catchment. The 2019 land use/cover scenario was then used to simulate the impact of the change in land cover on river discharge without changing the other SWAT input data (soils, slope and weather). The major assumption of this study is that the calibrated parameter set is still valid under changing land use and climatic conditions. The potential combined effect of land use and climate change on river discharge was evaluated using scenarios derived from a suite of 3 GCM models under the RCPs 4.5, 6.0 and 8.5 Greenhouse Gas Emission scenarios, which represent a wide range of simulated future (2030–2060) climate conditions (Table 2). Scenario 1 (S1) is the baseline with land use from 1990 and climate data from 1972–1990. Scenario 2 (S2) represented river discharge under land use change only. Scenario 3 (S3) was climate change only. Finally, scenario 4 (S4) represented the combined effect of land and climate change.

**Table 2.** Scenario results for the highest discharge at return periods 5, 25 and 100.

| Flow Index | Peak Discharge ($m^3s^{-1}$) | Difference | |
|---|---|---|---|
| | | **Value** | **%** |
| Q5 | | | |
| Baseline | 206.8 | | |
| LULC-Change (S2) | 207.5 | 0.74 | 0.36 |
| Climate Change (S3) | 212.2 | 5.36 | 2.59 |
| Combined Change(S4) | 213.7 | 6.91 | 3.34 |
| Q25 | | | |
| Baseline | 294.1 | | |
| LULC-Change (S2) | 295.1 | 0.92 | 0.31 |
| Climate Change (S3) | 300.2 | 6.09 | 2.07 |
| Combined Change (S4) | 310.2 | 16.08 | 5.47 |
| Q100 | | | |
| Baseline | 366.2 | | |
| LULC-Change (S2) | 367.3 | 1.06 | 0.29 |
| Climate Change (S3) | 390.2 | 24.01 | 6.56 |
| Combined Change (S4) | 400.4 | 34.13 | 9.32 |

To quantify the impacts of land use and climate change on river discharge in the Simiyu catchment for the years 1990 to 2019, the four scenarios (S1) to (S4) were used to run the calibrated SWAT model, and their outputs were compared.

For future simulation of climate change impacts to discharge (2030–2060), only S1 and S3 were considered since no predictions were made on future land use.

## 3. Results

### 3.1. Sensitivity Analysis, Model Calibration and Validation

The parameters' rankings remained relatively stable with and without observed data. The significant difference was observed in CH_N2, Alpha_BF, SURLAG, and CH_K2, which consequently provides an insight into the most sensitive parameters. The top 20 parameters were ranked based on sensitivity analysis (Table 3), and the 16 most sensitive parameters (Table 4) were used for calibration.

The most sensitive parameter Cn2 has a high optimal value, which denotes a low infiltration capacity. The NSE, PBIAS and RSR had values of 0.42, +1.5 and 0.74, respectively, before calibration, which were deemed good/satisfactory for NSE and PBIAS and unsatisfactory for RSR according to Moriasi et al., [46], while the *p*-factor was 39% and *r*–factor 65%. However, the model efficiency improved to 0.57, −0.70 and 0.53 for NSE, PBIAS and RSR, respectively, for the monthly time step. The *p* factor was 47%, *r* factor was 57% and NSE was 0.36 during the validation period (1988–1992) for the daily time step.

**Table 3.** Sensitivity analysis parameter ranking and fitted values after calibration.

| Parameter | Description | Rank | |
|---|---|---|---|
| | | With Obs | Without Obs |
| Cn2 | Curve number for moisture condition 11 | 1 | 1 |
| Esco | Soil evaporation compensation factor | 2 | 2 |
| Ch_K2 | Efficient hydraulic conductivity in the main channel alluvium (mm/hr) | 3 | 13 |
| Surlag | Surface runoff lag coefficient | 4 | 16 |
| Alpha_Bf | Baseflow alpha factor | 5 | 12 |
| Ch_N2 | Manning *n* value for the main channel | 6 | 15 |
| Canmx | Maximum canopy index | 7 | 5 |
| Blai | Maximum potential leaf area index | 8 | 8 |
| Sol_Awc | Available water capacity of the soil layer | 9 | 4 |
| Sol_Z | Soil depth(mm) | 10 | 3 |
| Slope | Average slope steepness(mm) | 11 | 7 |
| Revapmn | Threshold depth of water in the shallow aquifer for revap or percolation to deep aquifer to occur | 12 | 10 |
| Sol_K | Saturated hydraulic conductivity (mm/h) | 13 | 6 |
| Gw_Revap | Ground water "revap" coefficient | 14 | 11 |
| Gwqmn | Threshold depth of water in the shallow aquifer required for return flow to occur | 15 | 9 |
| Epco | Plant uptake compensation factor | 16 | 14 |
| Gw_Delay | Ground water delay | 17 | 18 |
| Biomix | Biological mixing coefficient | 18 | 19 |
| Slsubbsn | Average slope length | 19 | 20 |

**Table 4.** Sensitive parameters used in model calibration.

| Sn | Parameter Name | Fitted Value | Min Value | Max Value |
|---|---|---|---|---|
| 1 | R__CN2.mgt | −0.13 | −0.13 | −0.11 |
| 2 | V__ALPHA_BF.gw | 0.64 | 0.60 | 0.66 |
| 3 | V__GW_DELAY.gw | 413.55 | 350.39 | 492.33 |
| 4 | V__GWQMN.gw | 1080.60 | 813.15 | 1092.62 |
| 5 | V__GW_REVAP.gw | 0.17 | 0.16 | 0.19 |
| 6 | V__RCHRG_DP.gw | 0.27 | 0.23 | 0.31 |
| 7 | V__SURLAG.bsn | 8.94 | 8.70 | 9.56 |
| 8 | V__CH_N2.rte | 0.16 | 0.16 | 0.17 |
| 9 | V__CH_K2.rte | 71.18 | 59.67 | 87.27 |
| 10 | V__ESCO.hru | 0.37 | 0.36 | 0.38 |
| 11 | V__CANMX.hru | 0.04 | 0.04 | 1.67 |
| 12 | R__HRU_SLP.hru | 0.46 | 0.40 | 0.48 |
| 13 | R__SOL_AWC(..).sol | −0.12 | −0.13 | -0.08 |
| 14 | R__SOL_K(..).sol | 0.51 | 0.40 | 0.51 |
| 15 | R__SLSUBBSN.hru | 0.13 | 0.12 | 0.20 |
| 16 | V__EPCO.hru | 0.79 | 0.78 | 0.83 |

*3.2. Impacts of the Current Land Use and Climate Change on the River Discharge*

The World Meteorological Organization's recommendation to use the 18-year-period as a baseline was adopted to represent the baseline for land use and climate data from 1972 to 1990 [69]. The land use change (S2) was attributed to an increased peak discharge of 0.32% from the baseline. The climate-change-only (S3) scenario showed an increase in peak discharge by 3.72%. The combined impacts (S4) estimated an increase in peak discharge by 6.04%, indicating a synergistic impact of land use and climate change. For the discharges at a return period (T) of 25 years, the baseline discharge (S1) was 294.1 $m^3s^{-1}$ and S2 increased the discharge by 0.31%, S3 by 2.07% and S4 by 5.47%. The T = 100 years discharge events demonstrated that S3 had a higher increase (6.55%) compared to S2 (0.29%), while S4 caused an increase of 9.32% (Table 2).

### 3.3. Future Climate Changes under RCPs 4.5, 6.0 and 8.5

3.3.1. Projected Future Temperature and Precipitation Changes

The downscaled ensemble of GCM (ACCESS1.3, MIROC5 and CNRM.CM5) models predicted the change in the mean annual temperature from 21.8 °C in 1990 to about 22.2 °C at the end of 2019, an increase of about 0.4 °C over the past 30 years. The temperatures of the period between 2030 and 2060 under RCPs 4.5, 6.0 and 8.5 were predicted to increase by 0.6 °C from 22.6 °C in 2030 to 23.2 °C in 2060 in response to increasing greenhouse gas (GHG) concentrations and a reduction in the rainfall amount (Figures 4 and 5). The future temperature increases are mostly concentrated from December to August, while in September, October and November the future temperature is predicted to decrease very slightly.

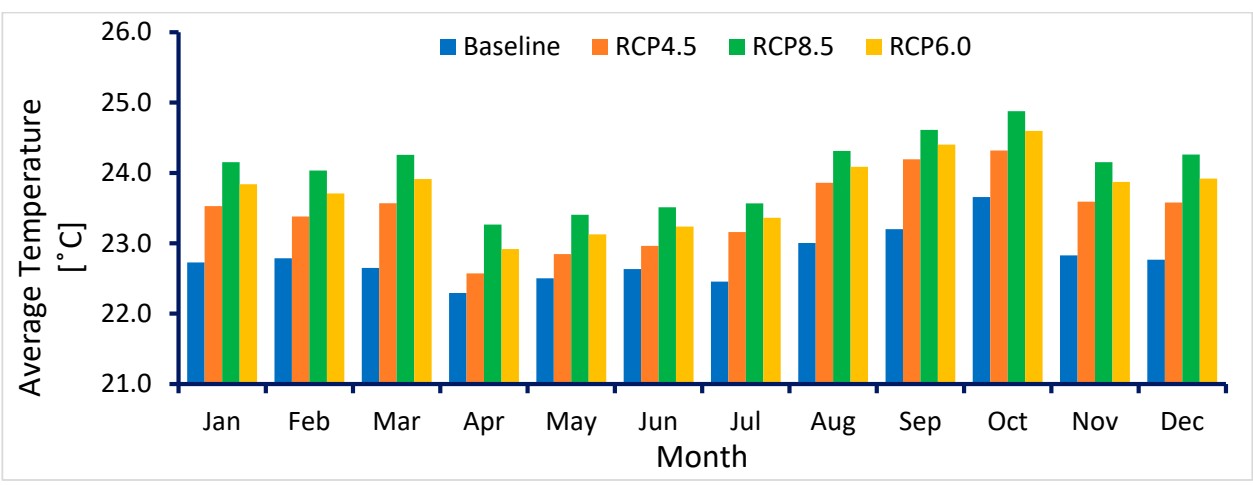

**Figure 4.** Comparison between the current 1990–2019 (baseline) and future temperature (2030–2060) at Simiyu catchment.

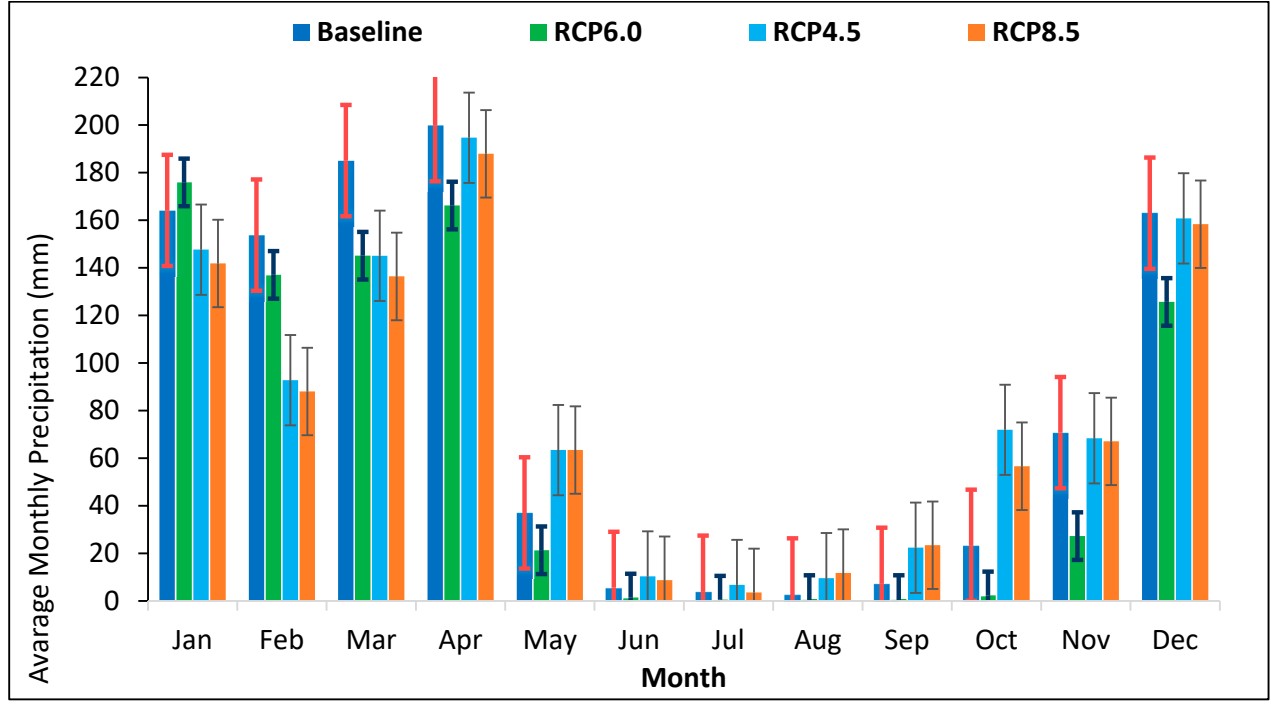

**Figure 5.** Comparison between the current 1990–2019 (baseline) and future rainfall (2030–2060) in Simiyu catchment.

### 3.3.2. Projected Future Annual and Seasonal River Discharge

The results showed that the river discharge varies across the months following the spatial and interannual variability in rainfall across the two wet seasons with the long rains from March to May, short rains in October to December and two intermediate dry seasons [29]. In general, the discharge increased from October through January, decreased afterwards up to March and then started increasing again until May, reaching its peak in April (Figure 6). These large differences between seasons are driven by the seasonal differences in precipitation and temperature changes. Mean monthly discharge forecasts during 2030–2060 under RCPs 4.5, 6.0 and 8.5 showed the maximum in April, which is consistent with the timing of the observed mean monthly discharge in the current period (1990–2019). In addition, predicted average monthly discharges are lower in the months from June to September, which is in line with the historical data. According to the model, the annual average river discharge of the Simiyu will significantly decrease (Figure 6) due to projected decreasing rainfall and increasing temperature in the catchment. Under RCPs 4.5, 6.0 and 8.5, the average annual river discharge decreased from 5.7 m$^3$/s in 1990–2019 to 4.0 m$^3$/s in 2030–2060, which is equivalent to a 29% decline (Table 2).

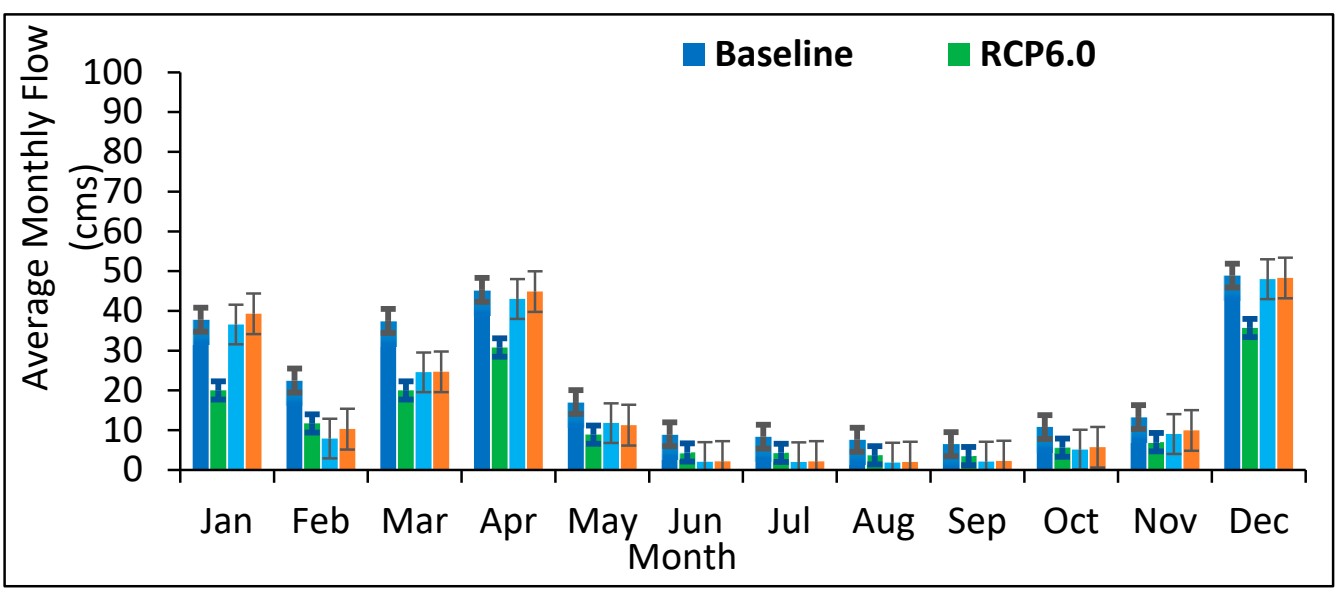

**Figure 6.** Comparison of the current and future flows.

## 4. Discussion

### 4.1. Sensitivity Analysis, Model Calibration and Validation

The sensitive parameters show that the catchment seems to be governed more strongly by surface runoff parameters compared to base flow parameters, which is expected in the semi-arid tropical systems [46]. A poor *p*-factor in validation is attributed to uncertainties in the data and the failure of the model to capture some hydrological catchment processes typical for semi-arid tropical catchments. Moreover, the model simulations for the daily time step slightly underestimated the peak river discharges (Figures 7 and 8). Similar results were obtained by [70] who validated the poor SWAT performance on accurate estimations of the daily catchment rainfall and lack of spatial distribution in climate data. Nonetheless, our model performed reasonably well since the efficiency performance of the independent parameters RSR, NSE and PBIAS attained during the calibration and validation period were within the recommended values (NSE > 0.5, PBIAS < ±25% and RSR < 0.7) [46]. In order to study the impact of future climate data on river discharge, the calibrated models were combined with downscaled future climate data.

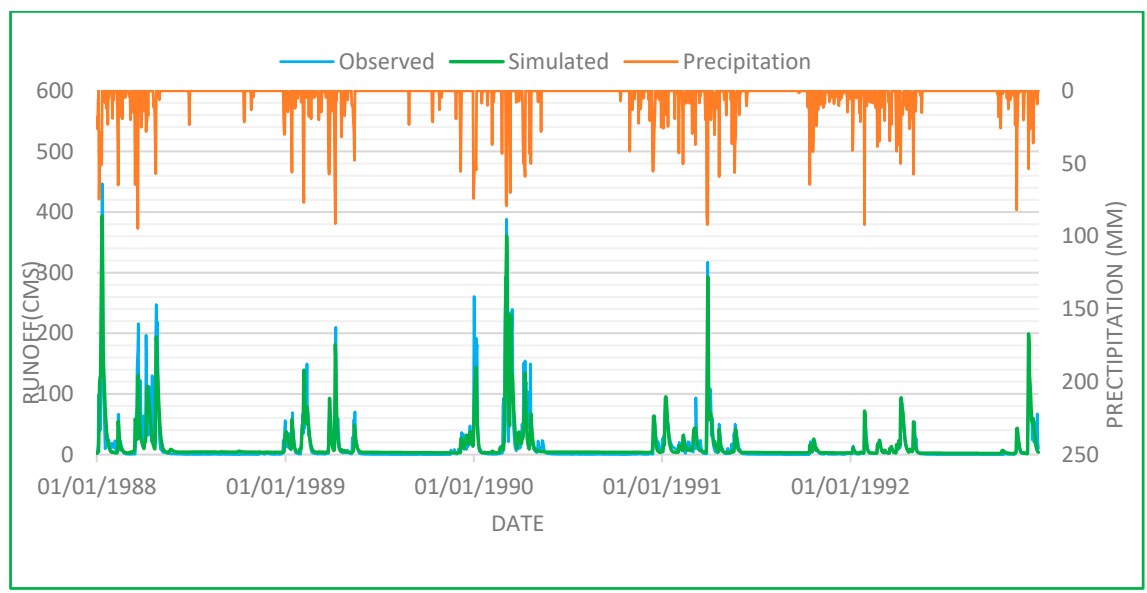

**Figure 7.** Observed and simulated daily flows for the calibration period (1988–1992).

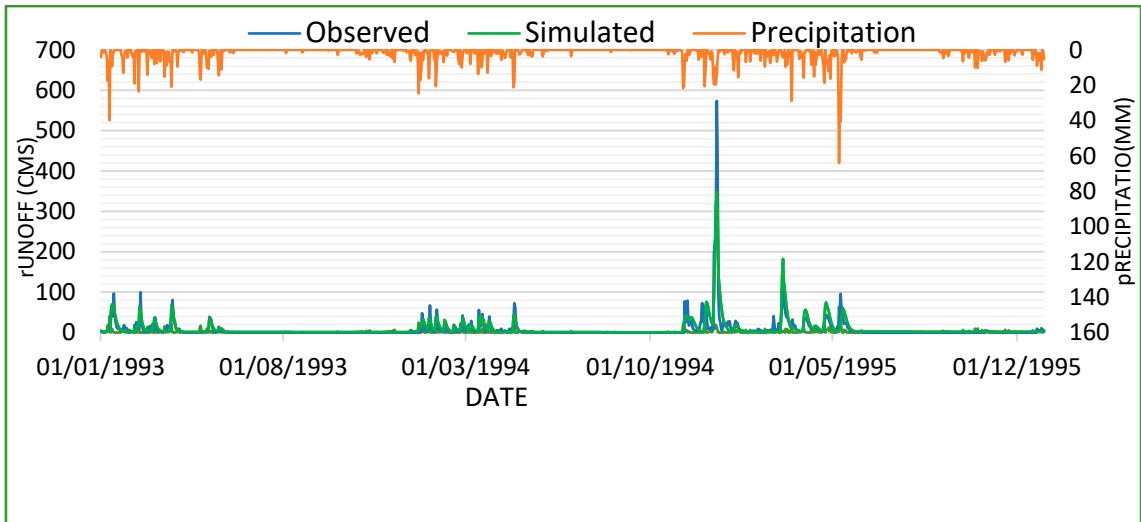

**Figure 8.** Observed and simulated daily flows for the validation period (1993–1996).

The NSE and RSR goodness-of-fit evaluation revealed that the simulated flow fitted the observed flow best as indicated in Figures 7 and 8. It was also observed that precipitation positively correlates with simulated and observed flows (Figures 7 and 8). The average performance of the model when simulating specific peaks was probably due to errors in estimation of daily catchment rainfall, spatial variability in rainfall in the catchment, and inadequate representation of Hortonian overland flow in the model. Other reasons might be water abstractions for domestic and socioeconomic activities (e.g., irrigation practices and mineral processing) that are not included in the modulation. Earlier studies reported that the SWAT model tends to underestimate the discharge peaks [71].

*4.2. Projected Future Climatological Change Variables under RCPs 4.5, 6.0 and 8.5*

According to IPCC (2021), the projection for high to moderate emission scenarios shows that by the middle and end of this century, the maximum and minimum temperatures over equatorial East Africa will rise and that there will be warmer days compared to the current situation. In the Simiyu catchment, this trend has begun to emerge because the first nine months of the year showed an increasing trend and the final three months

showed a very slight decreasing trend. Increasing soil evaporation and plant transpiration due to increasing rising temperatures as a result of climate change may have an impact on the soil–water balance, which would then have an impact on crop growth and agricultural productivity [62]. According to these results, the rainfall will be declining over time (Figure 5). The ensemble of GCMs (ACCESS1.3, CMCC.CM, MIROC5 and CNRM.CM5) under RCPs 4.5, 6.0 and 8.5 predicted that from 2030 to 2060, rainfall over the Simiyu River catchment will be reduced by 7.75% on average, according to the climate change models that are currently available.

The predicted low discharge in the dry season between June to September under all the land use and climate change scenarios is due to low rainfall and warm temperatures that lead to higher evapotranspiration, which often decreases runoff and discharge. The single peak discharges are frequently linked to vastly heavy rainfall events and likely occur on timescales smaller than the daily time step of the simulation period (Figure 6). The catchment is likely to experience longer and more pronounced droughts in the future, which has also been highlighted by the IPCC (2021) in the East Africa region. However, changes in precipitation are also predicted to drive changes in flood regimes [72]. The ensemble GCM models (ACCESS1.3, MIROC5 and CNRM.CM5) under RCPs 4.5, 6.0 and 8.5 predict higher incidences of extreme discharge as shown in Table 5 and in the flow duration curve at Ndagalu gauging station (Figure 9), which indicates frequent flood occurrence in the future (2030–2060) compared to the current period (1990–2019), with extreme discharges of 451.3 $m^3s^{-1}$ and 232.8 $m^3s^{-1}$ at exceedance probabilities of 0.01% and 99.99%, respectively. This might be attributed to mutual effects of increased land use and climate change. Intense precipitation events are predicted to produce a larger fraction of runoff that increases the probability of Hortonian overland flow in the catchment. The higher incidence of high-intensity rainfall is thus predicted to cause both more intense flood and drought events. In addition, extreme events will occur more frequently and also intensify, with large discharge events (floods) generally increasing in magnitude and frequency in the wet months March and April and low flows (droughts) occurring in the dry months of June to September in particular (Figure 6). These findings show less extreme discharge events [73] for the period of 1990–2019 compared to the future projected river discharge (2030–2060), which shows increased extreme events in the catchment because of higher intensity rainfall events following dry periods due to land degradation and short spells of heavy rainfall and prolonged dry spells (Figure 5).

The present climate-change-only scenario (1990–2019) caused the highest increase in discharge at different return periods compared to the land use change only scenario. The mutual impacts of climate and land use changes showed a disproportional increase in discharge compared to the single contributions, which indicates synergistic effects of land cover and climate change. However, the contribution ratio of climate change was larger than for land use change. The model simulations under projected climate change (2030–2060) showed a significant decrease in the discharge at different return periods. The dominant factor for the decrease in discharge was the decrease in precipitation. The observation that discharge dynamics were mainly controlled by precipitation variability rather than temperature is realistic due to the smaller relative differences in temperature between the seasons and future climate scenarios compared to those in rainfall. These results are in line with the IPCC [74] projected impacts of climate change in developing countries. According to the IPCC (2001), developing countries are the most vulnerable to climate change and climate variability [54]. As such, the availability and variability of fresh water will be greatly impacted by climate change in response to global warming, thus significantly affecting the economy of developing countries that heavily depend on agricultural production [75,76]. Most likely, the anticipated changes in the rainfall patterns and intensity in the river discharges will have an impact on crop growth and put farmers at risk from floods, soil erosion and drought. These findings underline the vulnerability of river discharge to rising greenhouse gas concentrations resulting from alterations in the climate system and stress the significance of global emission reduction strategies and

measures to protect future water resources. Since 80% of the Tanzania population is largely dependent on agriculture [77,78], we anticipate further increases in water demand as a result of population increases. It is therefore important to establish adaptation and mitigation measures to minimize the impacts of climate change on water resources.

**Table 5.** Comparison of the extreme discharges at exceedance probabilities of Simiyu catchment at Ndagalu gauging station (5D1).

| Exceedance Probability (%) | Extreme Discharge (m³/s) (Baseline) | Extreme Discharge (m³/s) (RCP4.5) | Extreme Discharge (m³/s) (RCP6.0) | Extreme Discharge (m³/s) (RCP8.5) |
|---|---|---|---|---|
| 0.01 | 379.6 | 466.0 | 451.3 | 413.2 |
| 1 | 232.8 | 276.9 | 214.3 | 239.7 |
| 5 | 102.2 | 136.5 | 72.3 | 129.3 |
| 10 | 34.2 | 65.5 | 32. | 62.6 |
| 20 | 14.8 | 16.8 | 10.9 | 16.8 |
| 25 | 10.6 | 10.9 | 8.0 | 11.1 |
| 50 | 6.5 | 3.9 | 5.3 | 3.8 |
| 75 | 5.1 | 2.8 | 3.9 | 3.0 |
| 90 | 4.0 | 2.0 | 2.8 | 2.6 |
| 95 | 3.2 | 1.9 | 1.9 | 2.7 |
| 99 | 1.4 | 0.1 | 0.2 | 0.1 |
| 100 | 1.2 | 0.00 | 0.0 | 0.00 |

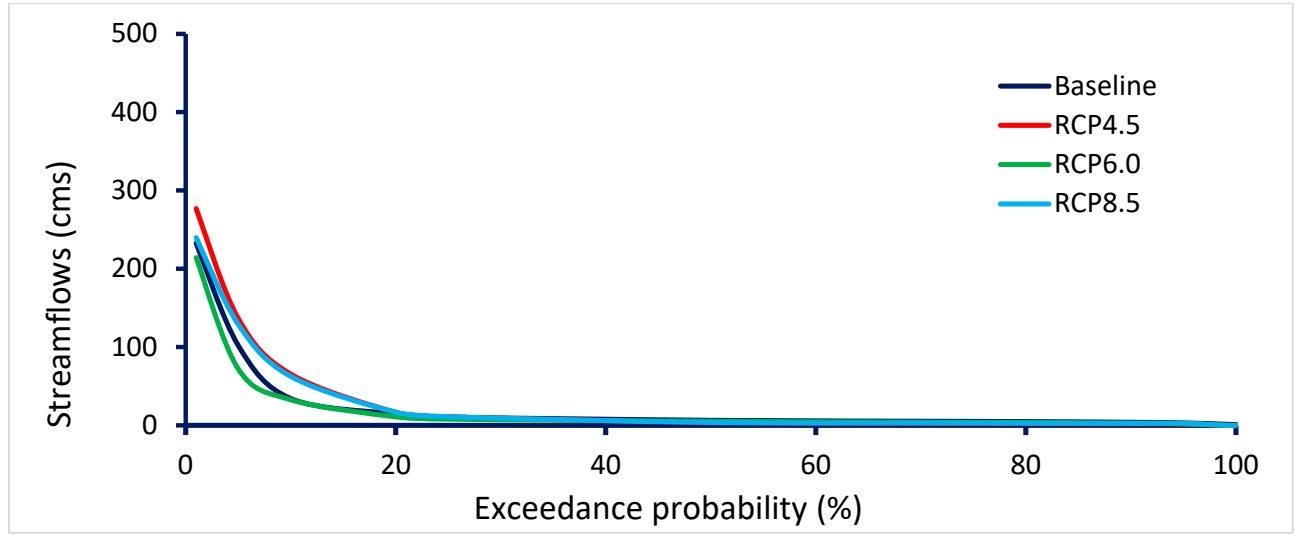

**Figure 9.** Flow duration curves at Ndagalu gauging Station (5D1) for current and future flows.

While the model performance was decent, there remained some challenges and weaknesses. The model's underestimations of high-flow events are the main cause of uncertainty and have an impact on the simulations of land use and the climate. The most significant source of uncertainty is due to the model underestimating high-flow events, which affects land use and climate simulations. This underestimation can be partly explained by the absence of enough gauged hydrometric and rainfall stations in the catchment to capture the high spatial and temporal variability in rainfall, thereby affecting simulated flow. Moreover, we simulated discharge on a daily scale, but rainfall in the study area often comes in high-intensity torrential rains. The high intensity of the rainfall in real life often passes the soil infiltration threshold, leading to Hortonian overland flow. However, in the daily setting, the model might assume that the rainfall is distributed evenly in the day and therefore predict that more of the rainfall infiltrates. In this context, stream discharge models in the East African region could be improved by rehabilitating non-operating rainfall stations and

collecting rainfall data on a higher spatial resolution. Discharge and rainfall monitoring should also aim to increase the resolution to hourly time steps for capturing high intensity rainfall events. Herein, future modelling exercises can follow suit and model stream discharge on hourly time steps, allowing a better representation of Hortonian overland flow. This will enhance the model's performance during calibration and hydrological simulations in upcoming studies. Choosing the GCM model(s) and defining the emission scenarios is also expected to have a major impact on future simulations [58–60]. This study used an ensemble of GCM models (ACCESS1.3, MIROC5 and CNRM.CM5) and three representative concentration pathways (RCPs 4.5, 6.0 and 8.5) to understand the impact of future climate change on the Simiyu river discharge (Figures 3 and 9). Different GCM models have shown discrepancies over regional climate change [79,80], e.g., due to differences in the spatial domains and predictor variables, downscaling with dynamic and statistical downscaling methods [79,81,82] or even within different statistical downscaling methods [83,84]. However, this was overcome by locally validating the models and selecting the one with the lowest inaccuracies for discharge modelling.

A further area of uncertainty relates to the hydrological model, which is used to interpret the impact of how future climate data will affect hydrological responses (e.g., influence on streamflow). The model structure, parameter uncertainty, and a lack of data all contribute to the hydrological model uncertainty [85]. However, this was partly overcome with local calibration using a semi-automatic calibration and uncertainty program, SWATCUP SUFI-2 [18]. While the parameters are calibrated for a specific climate and environment, it is not guaranteed that these settings remain optimal when changes in climate and land use occur. In spite of the model's limitations, this study made every effort to reduce the degree of uncertainty in the model's prediction to reasonably comprehend the feasible impact of climate change on the stream flow in Simiyu catchment. This study has revealed that the SWAT model is a robust method for predicting the hydrological response of semi-arid tropical catchments to changes in climate and land use. The model is thus an important tool for informing soil and water management strategies in these data-poor regions. Nevertheless, these findings should be hypothesized as best available simulations and not as empirical observations because of the temporally and spatially simplified representations of the catchment environment and hydrological processes.

## 5. Conclusions

In this study, the semi-distributed hydrological SWAT model was applied to evaluate the impact of current land use and climate change and assess the probable future effects of climate change in the near future (2030–2060) on Simiyu River discharge under RCPs 4.5, 6.0 and 8.5. The results indicated a significant increase in temperature of about 0.4 °C over the past 30 years (21.8 °C in 1990 to about 22.2 °C at the end of 2019). The climate model predicted the average annual temperature to increase by 1.4 °C in 2030 and by 2 °C in 2060, while the precipitation in the catchment is predicted to reduce by 7.8% in 2060 compared to the 1990–2019 baseline. These results imply that these changes will be accountable for alterations of the hydrological cycle by decreasing the Simiyu river discharge. The climate elasticities of the discharge revealed that the predicted changes in climate will result in a 29.0% total decrease in discharge in the catchment, which would result in a negative effect on the catchment's water resource availability. Moreover, the mutual effect of land use and climate change is predicted to have increased the chances of extreme discharges, more so than the singular effects of either climate change or land use change. Understanding the impact of climatic and land use changes on river discharge dynamics underpins constructive water management practices, particularly in vulnerable, arid and semi-arid environments of the Simiyu catchment. The findings from this research support the design of improved water resource management and adaptation strategies to climate change. Although the simulations of SWAT for the daily time step underestimated peak stream flow, the model has proved to be an appropriate tool for water flow prediction in large-scale catchments.

**Supplementary Materials:** The following supporting information can be downloaded at: https://www.mdpi.com/article/10.3390/earth4020020/s1.

**Author Contributions:** Conceptualization, R.J.S., M.W., K.M.M., J.N. and K.N.N.; data curation, R.J.S., A.I.A. and M.W.; formal analysis, R.J.S. and A.I.A., investigation, R.J.S.; methodology, R.J.S., A.I.A., M.W., K.M.M., J.N. and K.N.N.; project administration, R.J.S. and K.M.M.; resources, R.J.S. and A.I.A., software, R.J.S., A.I.A. and M.W., supervision, K.M.M., J.N. and K.N.N., validation, R.J.S., A.I.A. and, M.W., writing—original draft, R.J.S., writing—review and editing, M.W. and A.I.A. All authors have read and agreed to the published version of the manuscript.

**Funding:** This research was funded by the Lake Victoria Basin Water Board (LVBWB) which also covers the article processing charges.

**Data Availability Statement:** The data used to support the findings of this study are included in this article.

**Acknowledgments:** The first author would like to acknowledge the moral support and patient given by his family during this study.

**Conflicts of Interest:** The authors declare that there is no conflict of interest.

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
