# Peer review of "Assessing the Impacts of Land Use and Climate Changes on River Discharge towards Lake Victoria"

_2673-4834, doi:10.3390/earth4020020_

Round 1

Reviewer 1 Report

The manuscript depicts an interesting topic however some important adjustments need to be performed before going ahead in the pubblication phase and a strong re-arrangements of the text is necessary to be performed in order to improve the quality of the manuscript and its structure.

Firstly, I suggest you to improve the introduction section considering to depict climate change effects related to land use worldwide before focusing onto the study area and in particular enhancing the rolo of remote sensing in doing this. To do this part and help you I will provide you some useful references that I strongly advice to include in your manuscript

- https://doi.org/10.3390/geomatics3010012

- https://doi.org/10.1007/978-3-030-87934-1_12

- https://doi.org/10.1080/10106049.2021.1926552

Then in M&M consider to include the software adopted and how you performed the land cover which is the training set and how you validate them please report a confusion matrix and better explain this part

Then the graphs quality is very poor improve it.

In each maps report the resference system and the datum or EPSG!

Divide Results and Discussions into two separate sections. 

Discuss your foundings and scalability in other areas.

Conclusions are fine

If authors well follow the suggestion given I will certainly recommend your manuscript to be published on Earth

Reviewer 2 Report

The paper is a case study based on well known methodologies and approaches. The paper is prepared on good level.

There are many spelling errors in the text (spatila/spatial, hortonian/Hortonian, etc.). The text editing is needed.

Reviewer 3 Report

Comments:

Line 20. Climate changes. Probably “current climate”?

Lines 30-33. Please explain Q5 to Q100 abbreviations.

Line 48. Lake Victoria. Please specify “Northern Tanzania”.

Lines 49-52. I agree with this sentence, but the literature cited, especially 9, is very dated. I suggest deleting and inserting some more up-to-date citations, I am sure there are works in the literature referring to this area in Africa as well.

Line 103. “simiyu” In capital letters.

Lines 100-105. Please rewrite this part in a clearer version; for example, the first mentioned calibration, and then the aim of the study is presented.

Line 109. Figure 1. Please insert in the map values for low and high elevation.

Line 141. “to to”.

Lines 142-143. Slope map. Slop maps are not required in the SWAT model. Their use in the study needs clarification.

Lines 143-147. Land use/cover maps. Nothing is said about the data used for classification (testing and training) and about the final accuracy of these maps. Consider that if the classes are wrong, the impact on the hydrological balance can be considerable, if we consider that the main proportion is evapotranspiration, and this varies greatly between classes. Please provide more details regarding these maps.

Lines 147-151. Only precipitation, temp max and temp min were used. What about other variables? Please indicate that solar radiation, rel. humidity and wind speed, are not available; these variables are mandatory for Penman-Monteith ET calculation.

Line 159. Table 1. Rainfall stations. I think these are meteorological stations. Please correct. Furthermore, it refers to the “Percentage missing”. Please indicate which gap-filling measures were used for the missing data.

Line 154. Web URL. I cannot reach this site, it seems to be unavailable; please provide a valid address.

Line 184. HRU. The abbreviation is already presented in line 99. Please insert only abbr.

Lines 183-188. Nothing is stated about the evapotranspiration calculation method; please indicate this.

Lines 188-193. If daily river discharge data are available up to 31/12/1996, why does validation stop in 1992? Multi-decadal 30-year analysis is only a suggestion, if more data are present these should be used. Please clarify.

Line 194. Delete semicolon.

Lines 199-207. Please indicate the most sensitive parameters. These are useful for others interested in running the model in similar environments.

Lines 211-214. I have seen this paper but it is not clear why the 4 models selected should be suitable for Tanzania. Please clarify this passage better; also include more details of the chosen models as resolution, etc.

Line 217. Figure 3. It is unclear who produced this data and what exactly it represents; please clarify.

Lines 224-226. The choice of RCP 6.0 emission scenario as the most suitable because it assumes an increase in GHG emissions is highly subjective, scientifically misleading, and oriented towards telling climate change with a limited vision that does not consider other emission paths and scenarios. It is essential to analyse these scenarios taking into account others RCPs (4.5, 8.8)  comparing other GHG trends. Moreover, the cited reference (44) refers to India, and cannot say anything about possible scenarios for Tanzania.

In addition to the above comments, the authors are invited to resume modelling by extending the simulations by comprehensively considering the other RCPs.

Round 2

Reviewer 1 Report

The authors well followed the suggestion given the article is now suitable for pubblication.

Reviewer 3 Report

Please correct the section numbering; Discussions (4) start at paragraph 3.4. The same for conclusions (5).
